# Long-term risk of arrhythmias in patients with inflammatory bowel disease: A population-based, sibling-controlled cohort study

Jiangwei Sun [1]*, Bjorn Roelstraete[1], Emma Svennberg[2], Jonas Halfvarson [3], Johan Sundström [4,5], Anders Forss [1,6], Ola Olén [7,8,9], Jonas F. Ludvigsson [1,10,11]

1 Department of Medical Epidemiology and Biostatistics, Karolinska Institutet, Stockholm, Sweden, 2 Department of Medicine Huddinge, Karolinska University Hospital, Karolinska Institutet, Stockholm, Sweden, 3 Department of Gastroenterology, Faculty of Medicine and Health, Örebro University, Örebro, Sweden, 4 Department of Medical Sciences, Uppsala University, Uppsala, Sweden, 5 The George Institute for Global Health, University of New South Wales, Sydney, Australia, 6 Department of Gastroenterology, Dermatovenereology and Rheumatology, Gastroenterology unit, Karolinska University Hospital, Stockholm, Sweden, 7 Clinical Epidemiology Division, Department of Medicine Solna, Karolinska Institutet, Stockholm, Sweden, 8 Sachs' Children and Youth Hospital, Stockholm South General Hospital, Stockholm, Sweden, 9 Department of Clinical Science and Education Södersjukhuset, Karolinska Institutet, Stockholm, Sweden, 10 Department of Pediatrics, Örebro University Hospital, Örebro, Sweden, 11 Division of Digestive and Liver Disease, Department of Medicine, Columbia University Medical Center, New York, United States of America

* jiangwei.sun@ki.se

## Abstract

### Background

Although previous evidence has suggested an increased risk of cardiovascular disease (CVD) in patients with inflammatory bowel disease (IBD), its association with arrhythmias is inconclusive. In this study, we aimed to explore the long-term risk of arrhythmias in patients with IBD.

### Methods and findings

Through a nationwide histopathology cohort, we identified patients with biopsy-confirmed IBD in Sweden during 1969 to 2017, including Crohn's disease (CD: $n$ = 24,954; median age at diagnosis: 38.4 years; female: 52.2%), ulcerative colitis (UC: $n$ = 46,856; 42.1 years; 46.3%), and IBD-unclassified (IBD-U: $n$ = 12,067; 43.8 years; 49.6%), as well as their matched reference individuals and IBD-free full siblings. Outcomes included overall and specific arrhythmias (e.g., atrial fibrillation/flutter, bradyarrhythmias, other supraventricular arrhythmias, and ventricular arrhythmias/cardiac arrest). Flexible parametric survival models estimated hazard ratios (aHR) with 95% confidence intervals (95% CIs), after adjustment for birth year, sex, county of residence, calendar year, country of birth, educational attainment, number of healthcare visits, and cardiovascular-related comorbidities. Over a median of approximately 10 years of follow-up, 1,904 (7.6%) patients with CD, 4,154 (8.9%) patients with UC, and 990 (8.2%) patients with IBD-U developed arrhythmias, compared with 6.7%, 7.5%, and 6.0% in reference individuals, respectively. Compared with reference individuals, overall arrhythmias were increased in patients with CD [54.6 versus 46.1 per 10,000

---

**Data Availability Statement:** The data set cannot be shared directly under current legislation for data protection and must be requested directly from the

respective registry holders, Statistics Sweden (information@scb.se) and the Swedish National Board of Health and Welfare (registerservice@socialstyrelsen.se), after approval by the Swedish Ethical Review Authority.

**Funding:** This work was supported by FORTE (i.e., the Swedish Research Council for Health, Working Life and Welfare (https://forte.se/); Grant number 2016-00424 to JFL). The funder of the study had no role in study design, data collection, data analysis, data interpretation, or writing of the report.

**Competing interests:** ES has served as a speaker and/or advisory board member (all institutional grants) for Bayer, Bristol-Myers Squibb-Pfizer, Boehringer- Ingelheim, Johnson & Johnson, Merck Sharp & Dohme. JH served as speaker and/or advisory board member for AbbVie, Aqilion, BMS, Celgene, Celltrion, Dr Falk Pharma and the Falk Foundation, Ferring, Galapagos, Gilead, Hospira, Index Pharma, Janssen, MEDA, Medivir, MSD, Novartis, Pfizer, Prometheus Laboratories Inc., Sandoz, Shire, Takeda, Thermo Fisher Scientific, Tillotts Pharma, Vifor Pharma, UCB and received grant support from Janssen, MSD and Takeda. JS reports stock ownership in Anagram kommunikation AB and Symptoms Europe AB, outside the current study. AF has served as a speaker and advisory board member for Janssen. OO has been PI on projects at Karolinska Institutet financed by grants from Janssen, Pfizer, AbbVie, Takeda, Bristol Myer Squibb and Ferring, and Karolinska Institutet has received fees for lectures and participation on advisory boards from Janssen, Ferring, Galapagos, Bristol Myer Squibb, Takeda, and Pfizer. OO also reports grants from Pfizer, Janssen, Galapagos, and AbbVie in the context of a national safety monitoring programs. JFL has coordinated a study on behalf of the Swedish IBD quality register (SWIBREG). That study received funding from Janssen corporation. JFL has also received financial support from MSD developing a paper reviewing national healthcare registers in China. JFL is currently discussing potential research collaboration with Takeda. The other authors have declared that no competing interests exist.

**Abbreviations:** CD, Crohn's disease; CI, confidence interval; COPD, chronic obstructive pulmonary disease; CVD, cardiovascular disease; ESPRESSO, Epidemiology Strengthened by histoPathology Reports in Sweden; GI, gastrointestinal; HR, hazard ratio; IBD, inflammatory bowel disease; IBD-U, IBD-unclassified; ICD, International Classification of Diseases; IL, interleukin; IR, incidence rate; NPR, National Patient Register; PPV, positive predictive

person-years; aHR = 1.15 (95% CI [1.09, 1.21], $P < 0.001$)], patients with UC [64.7 versus 53.3 per 10,000 person-years; aHR = 1.14 (95% CI [1.10, 1.18], $P < 0.001$)], and patients with IBD-U [78.1 versus 53.5 per 10,000 person-years; aHR = 1.30 (95% CI [1.20, 1.41], $P < 0.001$)]. The increased risk persisted 25 years after diagnosis, corresponding to 1 extra arrhythmia case per 80 CD, 58 UC, and 29 IBD-U cases over the same period. Patients with IBD also had a significantly increased risk of specific arrhythmias, except for bradyarrhythmias. Sibling comparison analyses confirmed the main findings. Study limitations include lack of clinical data to define IBD activity, not considering the potential role of IBD medications and disease activity, and the potential residual confounding from unmeasured factors for arrhythmias.

## Conclusions

In this study, we observed that patients with IBD were at an increased risk of developing arrhythmias. The excess risk persisted even 25 years after IBD diagnosis. Our findings indicate a need for awareness of this excess risk among healthcare professionals.

## Author summary

### Why was this study done?

- Although previous studies have explored the associations between inflammatory bowel disease (IBD) and arrhythmias, earlier findings are inconclusive and unaddressed issues remain. For example, except for atrial fibrillation, data on the risk of other specific arrhythmias in patients with IBD is lacking.

- Until now, the long-term risk of overall and specific arrhythmias in patients with IBD remains unclear.

### What did the researchers do and find?

- In this population-based, sibling-controlled cohort study, we identified patients with biopsy-confirmed IBD in Sweden during 1969 to 2017, including Crohn's disease (CD) ($n = 24,954$), ulcerative colitis (UC) ($n = 46,856$), and IBD-unclassified (IBD-U) ($n = 12,067$), as well as their matched reference individuals and IBD-free full siblings.

- Patients with IBD were at a higher risk of developing overall arrhythmias than their matched reference individuals; the increased risk may persist even 25 years after IBD diagnosis, corresponding to 1 extra arrhythmia case per 80 CD, 58 UC, and 29 IBD-U cases over the same period.

- The risks of specific arrhythmias were also increased in patients with IBD, including atrial fibrillation/flutter, other supraventricular arrhythmias, and ventricular arrhythmias/cardiac arrest.

value; SNOMED, Systematized Nomenclature of
Medicine; TNF, tumor necrosis factor; UC,
ulcerative colitis.

### What do these findings mean?

- Healthcare professionals should be aware of the long-term increased risk of arrhythmias in patients with IBD.

- For those patients, a risk assessment of modifiable and established cardiovascular disease (CVD) risk factors could be considered.

- Study limitations include the absence of clinical data to define IBD activity, not considering the potential role of IBD medications and disease activity, and the potential residual confounding from unmeasured factors for arrhythmias.

## Introduction

Inflammatory bowel disease (IBD) is a chronic relapsing and remitting disease of the gastrointestinal (GI) tract, encompassing ulcerative colitis (UC), Crohn's disease (CD), and IBD-unclassified (IBD-U, subtype uncertainty) [1,2]. Although incidence of IBD has begun to stabilize or even decrease in some western countries [3], the burden of disease remains high or even increased due to high prevalence [4]. Because the peak incidence of IBD typically occurs in the second to fourth decade of life (the most productive period of adulthood) and there is currently no cure for IBD, it causes substantial productivity loss and has emerged as a big public health challenge worldwide [2,5]. In Sweden, around 0.8% of the Swedish population has been diagnosed with IBD [6,7].

Similar to immune-mediated inflammatory diseases, such as rheumatoid arthritis and psoriasis, IBD has been linked to increased cardiovascular disease (CVD) morbidity and mortality, including stroke, ischemic heart disease, and venous thromboembolism [8–13]. A potential systemic inflammation-driven etiology for cardiac arrhythmias has also been established via several mechanisms, including structural remodeling (e.g., accelerated atherosclerosis, myocardial injury, and fibrosis), electrical changes (e.g., causing inflammatory cardiac channelopathies), as well as indirect cardiac effects (e.g., changed central and peripheral sympathetic outflow) [14,15].

Although previous studies have explored the associations between IBD and arrhythmias [16–19] (summarized in Table A in S1 Appendix), the findings are inconclusive and there are some unaddressed issues. First, most of the studies have primarily focused on atrial fibrillation [16–19], with 1 study examining ventricular arrhythmias [18]. Evidence on the risk of overall and other specific arrhythmias in patients with IBD, including bradyarrhythmias, other supraventricular arrhythmias, and ventricular arrhythmias/cardiac arrest, is limited. Some studies of atrial fibrillation have been restricted to single hospitals [18] or included only prevalent cases [19], making it difficult to assess atrial fibrillation incidence and to extrapolate findings to a wider group of newly onset patients with IBD. In addition, a few studies applied a prospective study design to investigate the risk of atrial fibrillation in IBD [16,17], but suffered from short follow-up, detection bias, and surveillance bias. It therefore remains unclear if IBD is a risk factor for overall and specific arrhythmias, and whether such risk persists also years or decades after IBD diagnosis. A better understanding of their associations is imperative as arrhythmias have been suggested to lead to an increased risk of CVD [20] (the leading cause of mortality worldwide [21]), and until now no specific guidelines for CVD assessment and management in patients with IBD exist. Furthermore, evidence about whether any association with

arrhythmias varies by IBD phenotype is still scarce. Such knowledge would be beneficial for risk stratification, individualized prevention, and treatment in patients with IBD.

Taking advantage of the Swedish healthcare registers, we conducted a cohort study based on the nationwide histopathology cohort ESPRESSO (Epidemiology Strengthened by histoPathology Reports in Sweden) [22] to explore the risk of overall and specific arrhythmias in patients with biopsy-confirmed IBD. Drawing from the present knowledge about the role of systemic inflammation in the development of CVD and arrhythmia [14,15,23], we hypothesized that patients with IBD are at an increased risk of overall arrhythmias and specific types of arrhythmias. To account for differences in systemic inflammatory response, clinical features, and pathophysiology between IBD subtypes [1], we investigated the risk among patients with CD, UC, and IBD-U, separately.

## Methods

### Study design and participants

Sweden provides free tax-funded healthcare with equal access to all its residents [24]. ESPRESSO includes histopathology data from all 28 pathology departments in Sweden recorded between 1969 to 2017, including date on biopsy, topography, and morphology (recorded by the Swedish version of the Systematized Nomenclature of Medicine (SNOMED) system) [22]. We identified patients with IBD as those with at least 1 International Classification of Disease (ICD) code for IBD in the Swedish National Patient Register (NPR) and 1 biopsy record indicating IBD in the ESPRESSO (see Table B in S1 Appendix for ICD codes, SNOMED codes, and definitions of IBD subtypes) [3]. The NPR was established in 1964 and includes all inpatient care since 1987 and outpatient care since 2001 [25]. To minimize immortal time bias, date of IBD diagnosis (i.e., index date) was defined as the second of receiving first ICD or first biopsy codes. Such definition yields a high positive predictive value (PPV) of 95% for IBD diagnosis [26,27]. Data on CD location, UC extent, and perianal disease modifier were collected according to the Montreal classification [28]. CD location includes ileal (L1)/ileocolonic (L3)/unknown (LX) or colonic (L2). UC extent includes proctitis (E1)/left-sided colitis (E2), extensive colitis (E3) or extent not defined (EX) (see Table C in S1 Appendix for ICD codes). The prespecified analysis plan is presented in S1 Text.

### General population reference individuals and sibling comparators

For each patient with IBD, up to 5 reference individuals who were individually matched by birth year, sex, county of residence, and calendar year were randomly selected from the Total Population Register [29]. Full siblings to patients with IBD were identified through the Multi-Generation Register [30] to assess the influence of residual confounding from genetics and early environmental factors shared within families [31]. The reference individuals and full siblings had to be alive, living in Sweden, and free of IBD and arrhythmia when selected (i.e., index date).

### Follow-up and ascertainment of outcome

Study participants were linked to Swedish national healthcare registers by the unique personal identity number [32]. Individuals with previous arrhythmia (see Table D in S1 Appendix for ICD codes) before index date were excluded. Follow-up began at index date, with a virtually complete follow-up, until incident diagnosis of arrhythmias, emigration, death, or 31 December 2019, whichever occurred first. Some reference individuals or IBD-free full siblings were also censored when receiving a diagnosis of IBD during follow-up. Individuals with an

incident arrhythmia were identified through the NPR, considering both primary and secondary diagnoses, with a PPV of 97% for atrial fibrillation/flutter in a validation study [33]. The primary outcome was the incidence of overall arrhythmias, a composite outcome with ≥1 diagnosis of any individual outcome from the secondary outcomes. The secondary outcomes were incident atrial fibrillation/flutter, bradyarrhythmias, other supraventricular arrhythmias, and ventricular arrhythmias/cardiac arrest. If one individual was diagnosed with more than one of the secondary outcomes, this individual contributed to each outcome with the respective diagnosis date.

## Covariates

We retrieved information on country of birth (available from 1947 onward, Nordic (including Sweden, Denmark, Finland, Norway, and Iceland) or others) from the Total Population Register [29] and educational attainment (available from 1990 onward; 4 groups: 0 to 9 years, 10 to 12 years, ≥13 years, and "missing," a proxy for socioeconomic status) from the Swedish Longitudinal Integrated Database for Health Insurance and Labour Market Studies [34]. As a proxy for regular healthcare seeking behavior, number of healthcare visits were retrieved from the NPR (4 groups: 0, 1, 2 to 3, and ≥4,) which was defined as the number of specialized (non-primary care) outpatient visits or hospitalizations between 2 years and 6 months before index date. To avoid potential overadjustment, healthcare visits from 6 months before index date were not considered. Data on cardiovascular-related comorbidities preceding the index date were also collected from the NPR, including ischemic heart disease, heart failure, stroke, hypertension, diabetes, obesity, dyslipidemia, chronic kidney disease, and chronic obstructive pulmonary disease (COPD, only if the patient was diagnosed ≥40 years of age) (see Table E in S1 Appendix for ICD codes). Information regarding use of the cardiovascular-related medications preceding index date was collected from the Prescribed Drug Register [35] (data available from July 2005 onward): aspirin, non-aspirin anti-platelet medications, statins, non-statin lipid lowering medications, anticoagulation medications, antidiabetic medications, and antihypertensive agents (see Table F in S1 Appendix for the Anatomical Therapeutic Chemical codes).

## Statistical analyses

For all outcomes, we reported incidence rate (IR) and IR difference, together with 95% confidence intervals (CIs). To estimate the average and temporal patterns of hazard ratios (HRs) and 95% CIs for incident arrhythmias in relation to IBD, flexible parametric survival models were applied to allow the effect of IBD to vary over time (time-varying effect) rather than being constant [36] (see S2 Appendix for additional information about flexible parametric survival model). Time since index date was used as the underlying time scale. Standardized cumulative incidence and its differences of arrhythmia were also estimated using this approach [37]. We presented the cumulative incidence difference at 1 year, 5 years, 10 years, and 25 years after index date. In the population matched cohort, we conditioned the analyses on the matching variables (birth year, sex, county of residence, and calendar year) in model 1 and additionally adjusted for country of birth, educational attainment, number of healthcare visits, ischemic heart disease, heart failure, stroke, hypertension, diabetes, obesity, dyslipidemia, chronic kidney disease, and COPD in model 2.

## Subgroup and sensitivity analyses

We calculated the risk of overall arrhythmia by sex, age at index date (<18, 18 to 39, 40 to 59, and ≥60 years), calendar period at index date (1969 to 1989, 1990 to 1999, 2000 to 2009, and 2010 to 2019), educational attainment (0 to 9 years, 10 to 12 years, ≥13 years, and "missing"),

and number of healthcare visits (0, 1, 2 to 3, and ≥4). To examine the potential influence of disease phenotype, we further explored the associations by location for CD, by extent for UC according to the Montreal classification, and by occurrence of primary sclerosing cholangitis or other extraintestinal manifestations before the index date (see Table C in S1 Appendix for ICD codes) [28].

We conducted several sensitivity analyses to test the robustness of our results. First, due to lack of detailed data on smoking, we restricted the analysis to those without COPD (a proxy for heavy smoking) before the index date. Second, to assess the influence of cardiovascular-related comorbidities on the associations, we excluded those with cardiovascular-related comorbidities before the index date from the analysis. This analysis was added during the peer-review process as a non-prespecified analysis. Third, because information on educational attainment was available only from 1990 onward, and to assess its potential influence on results, we restricted the analysis to individuals with data available on educational attainment. Fourth, since the European prevalence of hypothyroidism and hyperthyroidism is about 5% [38] and 1% [39], respectively, and both Hashimoto's and Grave's disease are immune-mediated (similar to IBD), we further adjusted for autoimmune thyroid disease (see Table E in S1 Appendix for ICD codes) in the model to assess its influence on the associations. This analysis was added during the peer-review process as a non-prespecified analysis. Fifth, given that the Prescribed Drug Register became available in July 2005, we restricted the analysis to individuals with index date later than 1 January 2006 and constructed models that further adjusted for the abovementioned cardiovascular-related medications. Sixth, to assess the potential impact of detection bias (i.e., work up for 1 disease increases the chance of diagnosing the other) and surveillance bias (i.e., regular check-ups after IBD diagnosis increases chance of early detection of arrhythmias), we repeated the main analysis after excluding individuals with incident arrhythmias recorded within 1 year or within 3 years of follow-up after index date. Seventh, to assess the potential influence of confounding related to shared genetic or early environmental factors as well as healthcare-seeking behavior that may relate to IBD and arrhythmia risk within families, we compared patients with IBD with their IBD-free full siblings, after conditioning on family identifier and adjusting for birth year, sex, county of residence, calendar year, and all abovementioned covariates.

Data analyses were performed using SAS version 9.4 (SAS Institute, Cary, North Carolina, United States of America), Stata (version 16.1; Stata Corp LP, College Station, Texas, USA), and R version 3.6.0. A two-sided $P \leq 0.05$ was considered statistically significant. This study is reported as per the Strengthening the Reporting of Observational Studies in Epidemiology (STROBE) guideline (S1 Checklist).

### Ethics consideration

This study was approved by the Stockholm Ethics Review Board (2014/1287-31/4 and 2018/972-32). Individual informed consent was waived as the study was register based [40].

### Results

We identified 24,954 patients with CD (median age at index date: 38.4 years; female: 52.2%), 46,856 patients with UC (42.1 years; 46.3%), and 12,067 patients with IBD-U (43.8 years; 49.6%), together with their matched reference individuals (Table 1). Over two-thirds had been diagnosed after year 2000. Compared with reference individuals, patients with IBD had a higher number of healthcare visits and a higher prevalence of previous diseases, including ischemic heart disease, heart failure, stroke, hypertension, diabetes, obesity, dyslipidemia,

**Table 1. Characteristics of patients with inflammatory bowel disease and their matched reference individuals.**

| Characteristics | Patients with CD, n (%) | CD references, n (%) | Patients with UC, n (%) | UC references, n (%) | Patients with IBD-U, n (%) | IBD-U references, n (%) |
|---|---|---|---|---|---|---|
| N | 24,954 | 119,843 | 46,856 | 223,403 | 12,067 | 56,658 |
| Age at index date, years[a] | | | | | | |
| Mean ± SD | 40.7 ± 19.1 | 39.8 ± 18.6 | 43.7 ± 18.4 | 42.7 ± 17.9 | 44.7 ± 20.2 | 43.3 ± 19.6 |
| Median (IQR) | 38.4 (24.6, 55.3) | 37.4 (24.2, 54.0) | 42.1 (28.9, 57.6) | 41.1 (28.4, 56.1) | 43.8 (27.8, 61.0) | 42.2 (27.0, 59.0) |
| <18 | 2,859 (11.5) | 14,245 (11.9) | 3,044 (6.5) | 15,175 (6.8) | 1,163 (9.6) | 5,794 (10.2) |
| 18–39 | 10,213 (40.9) | 50,443 (42.1) | 18,600 (39.7) | 91,804 (41.1) | 4,181 (34.7) | 20,581 (36.3) |
| 40–59 | 7,157 (28.7) | 34,792 (29.0) | 15,035 (32.1) | 72,882 (32.6) | 3,548 (29.4) | 17,104 (30.2) |
| ≥60 | 4,725 (18.9) | 20,363 (17.0) | 10,177 (21.7) | 43,542 (19.5) | 3,175 (26.3) | 13,179 (23.3) |
| Female | 13,027 (52.2) | 62,800 (52.4) | 21,704 (46.3) | 104,046 (46.6) | 5,989 (49.6) | 28,213 (49.8) |
| Born in Nordic country[b] | 22,710 (91.0) | 106,616 (89.0) | 43,557 (93.0) | 199,055 (89.1) | 10,983 (91.0) | 49,628 (87.6) |
| Calendar period at index date[a] | | | | | | |
| 1969–1989 | 2,136 (8.6) | 10,550 (8.8) | 3,094 (6.6) | 15,183 (6.8) | 333 (2.8) | 1,635 (2.9) |
| 1990–1999 | 5,699 (22.8) | 27,720 (23.1) | 10,715 (22.9) | 51,661 (23.1) | 1,632 (13.5) | 7,828 (13.8) |
| 2000–2009 | 9,774 (39.2) | 46,759 (39.0) | 20,035 (42.8) | 95,365 (42.7) | 4,570 (37.9) | 21,584 (38.1) |
| 2010–2019 | 7,345 (29.4) | 34,814 (29.1) | 13,012 (27.8) | 61,194 (27.4) | 5,532 (45.8) | 25,611 (45.2) |
| Educational attainment, years | | | | | | |
| 0–9 | 5,684 (22.8) | 25,329 (21.1) | 10,034 (21.4) | 48,356 (21.7) | 2,710 (22.5) | 12,368 (21.8) |
| 10–12 | 9,523 (38.2) | 43,463 (36.3) | 18,882 (40.3) | 88,138 (39.5) | 4,736 (39.3) | 21,370 (37.7) |
| ≥13 | 5,061 (20.3) | 28,102 (23.5) | 11,825 (25.2) | 56,403 (25.3) | 2,936 (24.3) | 15,081 (26.6) |
| Missing | 4,686 (18.8) | 22,949 (19.2) | 6,115 (13.1) | 30,506 (13.7) | 1,685 (14.0) | 7,839 (13.8) |
| Number of healthcare visits[c] | | | | | | |
| 0 | 14,084 (56.4) | 90,742 (75.7) | 29,137 (62.2) | 169,743 (76.0) | 6,010 (49.8) | 39,561 (69.8) |
| 1 | 4,089 (16.4) | 14,523 (12.1) | 7,660 (16.4) | 26,806 (12.0) | 2,026 (16.8) | 7,714 (13.6) |
| 2–3 | 3,427 (13.7) | 8,892 (7.4) | 5,648 (12.1) | 16,744 (7.5) | 1,927 (16.0) | 5,503 (9.7) |
| ≥4 | 3,354 (13.4) | 5,686 (4.7) | 4,411 (9.4) | 10,110 (4.5) | 2,104 (17.4) | 3,880 (6.9) |
| Disease history before index date | | | | | | |
| Ischemic heart disease | 1,512 (6.1) | 4,452 (3.7) | 2,810 (6.0) | 9,257 (4.1) | 1,112 (9.2) | 2,947 (5.2) |
| Heart failure | 317 (1.3) | 638 (0.5) | 505 (1.1) | 1,379 (0.6) | 235 (2.0) | 447 (0.8) |
| Stroke | 840 (3.4) | 2,669 (2.2) | 1,419 (3.0) | 5,149 (2.3) | 675 (5.6) | 1,724 (3.0) |
| Hypertension | 2,366 (9.5) | 6,393 (5.3) | 4,051 (8.7) | 12,899 (5.8) | 1,895 (15.7) | 4,917 (8.7) |
| Diabetes | 1,332 (5.3) | 4,652 (3.9) | 2,571 (5.5) | 9,055 (4.1) | 1,117 (9.3) | 3,181 (5.6) |
| Obesity | 398 (1.6) | 1,192 (1.0) | 461 (1.0) | 2,180 (1.0) | 267 (2.2) | 761 (1.3) |
| Dyslipidemia | 641 (2.6) | 2,101 (1.8) | 1,265 (2.7) | 4,279 (1.9) | 547 (4.5) | 1,538 (2.7) |
| Chronic kidney disease | 138 (0.6) | 187 (0.2) | 205 (0.4) | 442 (0.2) | 157 (1.3) | 148 (0.3) |
| COPD | 512 (2.1) | 938 (0.8) | 780 (1.7) | 1,885 (0.8) | 393 (3.3) | 624 (1.1) |
| Montreal Classification CD at index date[d] | | | | | | |
| L1, L3/LX (Ileal, ileocolonic or location not defined) | 14,980 (60.0) | - | - | - | - | - |
| L2 (Colonic) | 3,644 (14.6) | - | - | - | - | - |
| Perianal | 1,305 (5.2) | - | - | - | - | - |
| Montreal Classification UC at index date[e] | | | | | | |
| E1/E2 (Proctitis, left-sided colitis) | - | - | 13,398 (28.6) | - | - | - |
| E3 (Extensive colitis) | - | - | 7,270 (15.5) | - | - | - |
| EX (Extent not defined) | - | - | 15,207 (32.5) | - | - | - |

(*Continued*)

**Table 1.** (Continued)

| Characteristics | Patients with CD, n (%) | CD references, n (%) | Patients with UC, n (%) | UC references, n (%) | Patients with IBD-U, n (%) | IBD-U references, n (%) |
|---|---|---|---|---|---|---|
| Extraintestinal manifestations at index date | | | | | | |
| Primary sclerosing cholangitis | 155 (0.6) | - | 734 (1.6) | - | 185 (1.5) | - |
| Other extraintestinal manifestations | 1,451 (5.8) | - | 1,752 (3.7) | - | 818 (6.8) | - |
| Follow-up time, years | | | | | | |
| Median (IQR) | 12.7 (7.0, 19.3) | 13.3 (7.4, 20.0) | 12.8 (7.2, 18.8) | 13.2 (7.5, 19.0) | 9.1 (5.1, 14.6) | 9.6 (5.6, 15.3) |
| 0–0.9 | 731 (2.9) | 2,108 (1.8) | 1,216 (2.6) | 3,917 (1.8) | 591 (4.9) | 1,560 (2.8) |
| 1–4.9 | 3,040 (12.2) | 13,670 (11.4) | 5,519 (11.8) | 25,073 (11.2) | 2,382 (19.7) | 10,261 (18.1) |
| 5–9.9 | 5,969 (23.9) | 28,080 (23.4) | 11,029 (23.5) | 52,142 (23.3) | 3,710 (30.8) | 17,758 (31.3) |
| 10–19.9 | 9,445 (37.9) | 45,969 (38.4) | 19,327 (41.3) | 92,566 (41.4) | 4,061 (33.7) | 19,977 (35.3) |
| ≥20 | 5,769 (23.1) | 30,016 (25.1) | 9,765 (20.8) | 49,705 (22.3) | 1,323 (11.0) | 7,102 (12.5) |

[a]Index date: date of IBD diagnosis for patients and date of selection for their matched reference individuals.

[b]Nordic country includes Sweden, Denmark, Finland, Norway, and Iceland.

[c]Defined as the number of healthcare visits between 2 years and 6 months before the index date.

[d]Not every patient with CD has location information.

[e]Not every patient with UC has extent information.

CD, Crohn's disease; COPD, chronic obstructive pulmonary disease; E, Extent; IBD-U, inflammatory bowel disease unclassified; IQR, interquartile range; L, location; SD, standard deviation; UC, ulcerative colitis.

chronic kidney disease, and COPD (Table 1). About 14.6% patients with CD were colonic and 15.5% patients with UC had extensive colitis (Table 1).

## Overall arrhythmias

During a median follow-up of around 10 years, 7.6% patients with CD ($n$ = 1,904; IR: 54.6 per 10,000 person-years), 8.9% patients with UC ($n$ = 4,154; IR: 64.7 per 10,000 person-years), and 8.2% patients with IBD-U ($n$ = 990; IR: 78.1 per 10,000 person-years) developed overall arrhythmias, compared with 6.7% (IR: 46.1 per 10,000 person-years), 7.5% (IR: 53.3 per 10,000 person-years), and 6.0% (IR: 53.5 per 10,000 person-years) in the reference individuals (Table 2). This corresponded to an IR difference of 8.5, 11.4, and 24.6 per 10,000 person-years, respectively. After multivariable adjustment, patients with IBD were at increased risks of overall arrhythmias, with an average aHR of 1.15 (95% CI [1.09, 1.21], $P$ < 0.001) for CD, 1.14 (95% CI [1.10, 1.18], $P$ < 0.001) for UC, and 1.30 (95% CI [1.20, 1.41], $P$ < 0.001) for IBD-U (Table 2). The elevated aHR decreased and reached a nadir around 6 to 7 years, but remained significantly increased 25 years after IBD diagnosis (Fig 1).

The cumulative incidence of overall arrhythmias was higher in exposed individuals (Fig 1), with a 10-year and 25-year cumulative incidence difference of 0.48% and 1.25% for CD, 0.48% and 1.73% for UC, and 1.24% and 3.48% for IBD-U, respectively (Table G in S1 Appendix). This corresponded to 1 extra estimated arrhythmia case per 208 and 80 patients with CD, 208 and 58 patients with UC, and 81 and 29 patients with IBD-U at 10 and 25 years after initial IBD diagnosis, respectively.

In stratified analyses for overall arrhythmias (Table H in S1 Appendix), although aHRs were significantly higher in patients with UC or IBD-U diagnosed at 18 to 39 years, compared with other age groups ($P_{\text{interaction}}$ = 0.039 and 0.008, respectively), IR and its difference was largest among those diagnosed ≥60 years. We observed higher aHR but lower IR among females than males, and both higher aHR and IR among those with ≥4 healthcare visits than

**Table 2. Incident arrhythmia in patients with inflammatory bowel disease and their matched reference individuals.**

| Outcomes | No. of events, n (%) | | IR (95% CI), per 10,000 Pys | | IR (95% CI) difference, per 10,000 Pys | HR (95% CI) | |
| --- | --- | --- | --- | --- | --- | --- | --- |
| | Patients | References | Patients | References | | Model 1[a] | Model 2[b] |
| CD | | | | | | | |
| Overall arrhythmias | 1,904 (7.6) | 8,033 (6.7) | 54.6 (52.2, 57.1) | 46.1 (45.1, 47.1) | 8.5 (5.8, 11.1) | 1.25 (1.19, 1.32) | 1.15 (1.09, 1.21) |
| Atrial fibrillation/flutter | 1,450 (5.8) | 6,310 (5.3) | 41.3 (39.2, 43.5) | 36.0 (35.2, 36.9) | 5.2 (2.9, 7.5) | 1.21 (1.14, 1.29) | 1.12 (1.05, 1.20) |
| Bradyarrhythmias | 268 (1.1) | 1,116 (0.9) | 7.5 (6.6, 8.5) | 6.3 (5.9, 6.7) | 1.2 (0.3, 2.2) | 1.28 (1.11, 1.48) | 1.10 (0.95, 1.28) |
| Other supraventricular arrhythmias | 215 (0.9) | 763 (0.6) | 6.0 (5.3, 6.9) | 4.3 (4.0, 4.6) | 1.7 (0.9, 2.6) | 1.43 (1.22, 1.67) | 1.35 (1.15, 1.58) |
| Cardiac arrest and ventricular arrhythmias | 324 (1.3) | 1,210 (1.0) | 9.1 (8.1, 10.1) | 6.8 (6.4, 7.2) | 2.3 (1.2, 3.3) | 1.40 (1.23, 1.60) | 1.24 (1.08, 1.43) |
| UC | | | | | | | |
| Overall arrhythmias | 4,154 (8.9) | 16,777 (7.5) | 64.7 (62.8, 66.7) | 53.3 (52.5, 54.1) | 11.4 (9.3, 13.5) | 1.21 (1.17, 1.25) | 1.14 (1.10, 1.18) |
| Atrial fibrillation/flutter | 3,261 (7.0) | 13,300 (6.0) | 50.4 (48.7, 52.2) | 42.0 (41.3, 42.8) | 8.4 (6.5, 10.3) | 1.18 (1.13, 1.23) | 1.12 (1.07, 1.17) |
| Bradyarrhythmias | 523 (1.1) | 2327 (1.0) | 7.9 (7.3, 8.6) | 7.2 (6.9, 7.5) | 0.7 (−0.0, 1.4) | 1.09 (0.98, 1.20) | 1.03 (0.93, 1.14) |
| Other supraventricular arrhythmias | 393 (0.8) | 1433 (0.6) | 6.0 (5.4, 6.6) | 4.4 (4.2, 4.7) | 1.5 (0.9, 2.1) | 1.34 (1.19, 1.50) | 1.31 (1.16, 1.47) |
| Cardiac arrest and ventricular arrhythmias | 692 (1.5) | 2,440 (1.1) | 10.5 (9.7, 11.3) | 7.6 (7.3, 7.9) | 2.9 (2.1, 3.7) | 1.38 (1.26, 1.51) | 1.25 (1.14, 1.37) |
| IBD-U | | | | | | | |
| Overall arrhythmias | 990 (8.2) | 3,373 (6.0) | 78.1 (73.3, 83.1) | 53.5 (51.7, 55.3) | 24.6 (19.4, 29.8) | 1.46 (1.35, 1.58) | 1.30 (1.20, 1.41) |
| Atrial fibrillation/flutter | 786 (6.5) | 2,638 (4.7) | 61.5 (57.3, 66.0) | 41.6 (40.0, 43.2) | 19.9 (15.4, 24.5) | 1.50 (1.37, 1.63) | 1.33 (1.22, 1.46) |
| Bradyarrhythmias | 119 (1.0) | 442 (0.8) | 9.1 (7.5, 10.9) | 6.9 (6.2, 7.5) | 2.2 (0.5, 4.0) | 1.31 (1.05, 1.63) | 1.21 (0.96, 1.53) |
| Other supraventricular arrhythmias | 87 (0.7) | 280 (0.5) | 6.6 (5.3, 8.2) | 4.3 (3.9, 4.9) | 2.3 (0.8, 3.8) | 1.50 (1.17, 1.93) | 1.34 (1.03, 1.74) |
| Cardiac arrest and ventricular arrhythmias | 149 (1.2) | 475 (0.8) | 11.4 (9.6, 13.3) | 7.4 (6.7, 8.1) | 4.0 (2.1, 5.9) | 1.53 (1.25, 1.87) | 1.39 (1.12, 1.72) |

[a]Conditioned on the matching variables (birth year, sex, county of residence, and calendar year).

[b]Further adjusted for country of birth, educational attainment, number of healthcare visits, ischemic heart disease, heart failure, stroke, hypertension, diabetes, obesity, dyslipidemia, chronic kidney disease, and COPD.

CD, Crohn's disease; CI, confidence interval; COPD, chronic obstructive pulmonary disease; HR, hazard ratio; IBD-U, inflammatory bowel disease unclassified; IR, incidence rate; UC, ulcerative colitis; Pys, person-years.

other strata ($P_{\text{interaction}}$ for sex was 0.068, 0.111, and 0.983 for CD, UC, and IBD-U, respectively; $P_{\text{interaction}}$ for healthcare visits was 0.566, 0.849, and 0.098 for CD, UC, and IBD-U, respectively). In stratified analyses by disease phenotype (Table I in S1 Appendix), aHRs were higher in those with other extraintestinal manifestations (HR = 1.39 (95% CI [1.06, 1.83], $P < 0.001$) for CD, 1.36 (95% CI [1.08, 1.70], $P < 0.001$) for UC, and 1.89 (95% CI [1.26, 2.83], $P < 0.001$) for IBD-U). Adjusted HRs were also higher in patients with colonic CD (aHR = 1.20 (95% CI [1.02, 1.41], $P < 0.001$) and in patients with extensive colitis (aHR = 1.20 (95% CI [1.08, 1.33], $P < 0.001$), but $P_{\text{interaction}}$ was 0.439 and 0.557 for CD and UC, respectively.

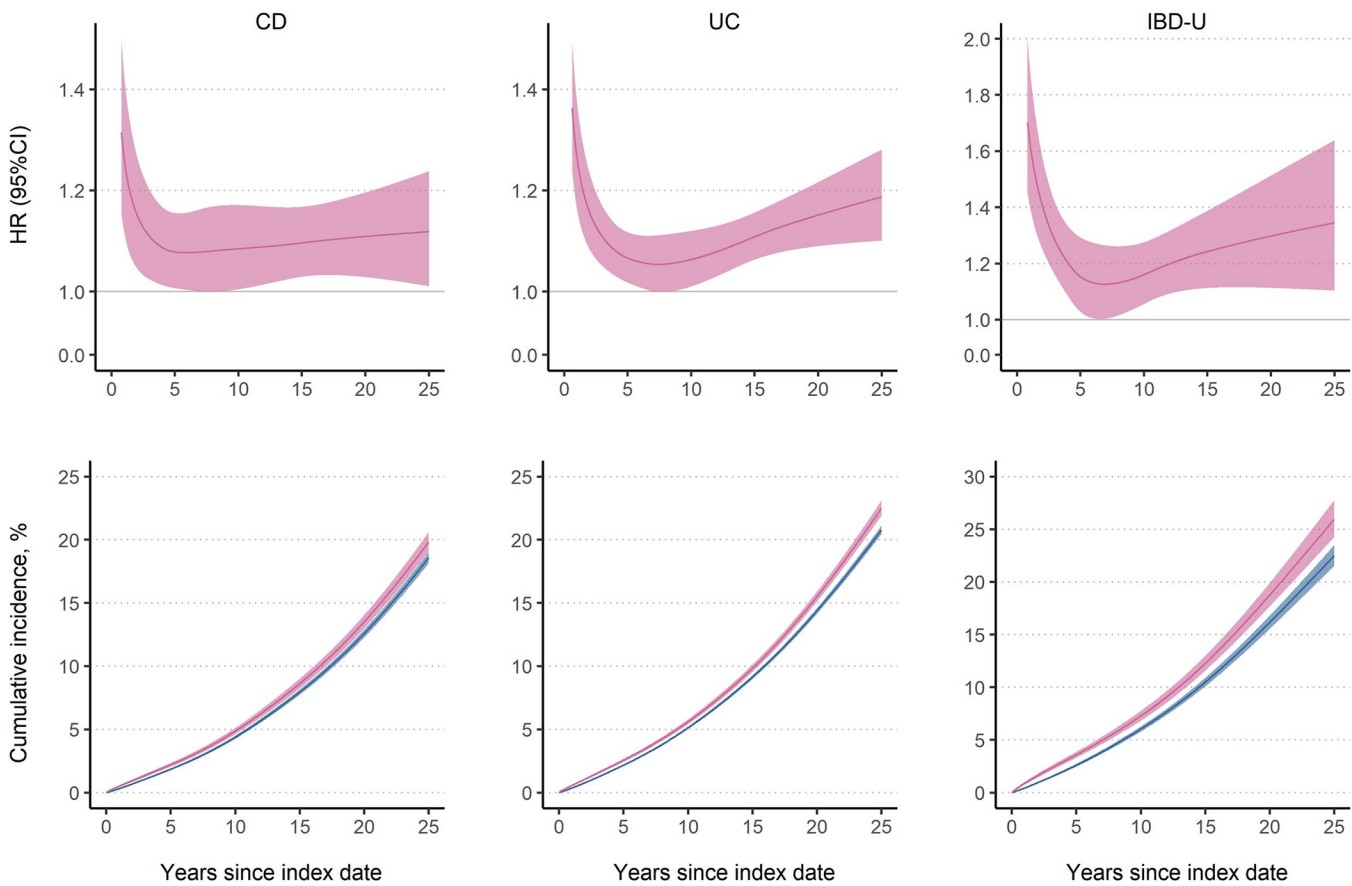

**Fig 1. HR and standardized cumulative incidence of overall arrhythmias.** Upper row: HR and 95% CI of overall arrhythmias, comparing patients with IBD with their reference individuals; lower row: standardized cumulative incidence and 95% CI of overall arrhythmias in patients with IBD (pink) and their reference individuals (blue). Both HR and standardized cumulative incidence were estimated from the flexible parametric survival model. CD, Crohn's disease; CI, confidence interval; HR, hazard ratio; IBD-U, inflammatory bowel disease unclassified; UC: ulcerative colitis.

## Secondary arrhythmia outcomes

Compared with the reference individuals, patients with CD, UC, and IBD-U had a significantly higher aHR of developing atrial fibrillation/flutter [aHR = 1.12 (95% CI [1.05, 1.20]), 1.12 (95% CI [1.07, 1.17]), and 1.33 (95% CI [1.22, 1.46]) for CD, UC, and IBD-U, respectively], other supraventricular arrhythmias [aHR = 1.35 (95% CI [1.15, 1.58]), 1.31 (95% CI [1.16, 1.47]), and 1.34 (95% CI [1.03, 1.74])], and ventricular arrhythmias/cardiac arrest [aHR = 1.24 (95% CI [1.08, 1.43]), 1.25 (95% CI [1.14, 1.37]), and 1.39 (95% CI [1.12, 1.72])], but there was no evidence of a significant association between IBD and bradyarrhythmias (Table 2, Fig A in S1 Appendix). Correspondingly, we observed gradually increased cumulative incidence difference over time between patients with IBD and the reference individuals for atrial fibrillation/flutter, other supraventricular arrhythmias, and ventricular arrhythmias/cardiac arrest (Fig B in S1 Appendix).

In stratified analyses by disease phenotype (Table J in S1 Appendix), we observed significantly increased risk of some specific arrhythmias such as atrial fibrillation/flutter in CD [aHR = 1.52 (95% CI [1.10, 2.10])] or IBD-U [aHR = 2.22 (95% CI [1.40, 3.52])] patients with other extraintestinal manifestations, other supraventricular arrhythmias in patients with CD and a perianal disease [aHR = 4.52 (95% CI [1.77, 11.53])] or in patients with UC and primary

sclerosing cholangitis [aHR = 4.99 (95% CI [1.28, 19.48])], and ventricular arrhythmias/cardiac arrest in patients with extensive colitis [aHR = 1.36 (95% CI [1.01, 1.83])].

## Sensitivity analyses

Robust results were observed across all sensitivity analyses after restricting the cohort to those without a diagnosis of COPD, to those without cardiovascular-related comorbidities, to those with data available on educational attainment, and to those with index date later than January 2006 as well as after further adjusting for autoimmune thyroid disease (Table K in S1 Appendix). After excluding individuals with incident arrhythmias recorded within 1 year of follow-up or within 3 years of follow-up (Table L in S1 Appendix), similar results were still noted for IBD and risk of arrhythmias.

## Sibling comparison

We identified 16,287 patients with CD, 30,556 patients with UC and 7,694 patients with IBD-U with ≥1 IBD-free full sibling alive at index date to address potential residual confounding from intra-familial factors relevant to IBD and arrhythmia (Table M in S1 Appendix). Patients with IBD were younger and more likely to have higher number of healthcare visits and higher prevalence of previous diseases than their IBD-free full siblings. Consistent with the primary analysis, patients with UC or IBD-U demonstrated significantly higher risk of developing overall arrhythmias [aHR = 1.18 (95% CI [1.09. 1.28], $P < 0.001$) for UC; aHR = 1.19 (95% CI [0.99, 1.42], $P = 0.058$) for IBD-U] and atrial fibrillation/flutter [aHR = 1.19 (95% CI [1.09, 1.30], $P < 0.001$) for UC; aHR = 1.24 (95% CI [1.01, 1.53], $P = 0.044$) for IBD-U, Table N in S1 Appendix], but not for patients with CD [aHR = 1.09 (95% CI [0.97, 1.22], $P = 0.142$) for overall arrhythmias; aHR = 1.07 (95% CI [0.93, 1.22], $P = 0.341$) for atrial fibrillation/flutter]. We also observed increased risk of other supraventricular arrhythmias in patients with CD [aHR = 1.39 (95% CI [1.04, 1.86], $P = 0.025$)] and ventricular arrhythmias/cardiac arrest in patients with UC [aHR = 1.27 (95% CI [1.04, 1.56], $P = 0.020$)].

## Discussion

In this nationwide population-based cohort study, we found that patients with IBD were at a higher risk of developing overall and specific arrhythmias than reference individuals and their IBD-free full siblings. The increased risk persisted over 25 years after IBD diagnosis and was higher among patients diagnosed at 18 to 39 years. No significant association was observed for bradyarrhythmias. The results were robust in a series of sensitivity analyses.

Our results of an increased risk of atrial fibrillation/flutter in patients with IBD (adjusted HRs ranged from 1.12 in CD and UC to 1.33 in IBD-U) are consistent with previous findings [16,17]. A Korean study [16] of 37,696 patients with IBD (mean age: 39.4 years; female: 39.0%) found an increased risk of non-valvular atrial fibrillation in patients with IBD [aHR = 1.36 (95% CI [1.20, 1.54])], with higher HRs in patients with CD, younger patients, and those with moderate-to-severe IBD (i.e., those receiving systemic corticosteroids in combination with 5-aminosalicylic acid, immunomodulators, or biological agents). In a Danish study [17] of 24,499 patients with IBD (mean age: 43.9 years; female: 53.9%), the authors reported an elevated overall-IBD-associated risk of atrial fibrillation, with an IR ratio of 1.26 (95% CI [1.16, 1.36]). The increased risk was primarily driven by the excess atrial fibrillation incidence during IBD flare and persistent activity period rather than remission period. Unlike those 2 studies, we used ICD codes to characterize disease phenotype and observed higher risk of overall arrhythmias in patients with colonic CD, in patients with extensive colitis, and in those with

other extraintestinal manifestations, which might be caused by the heavy inflammatory burden and severe disease activity. To the best of our knowledge, our study is the first one to date to explore the long-term risk of newly diagnosed specific arrhythmias (except for atrial fibrillation) in patients with IBD. Although statistical power is an issue in the subgroup analyses by Montreal Classification for specific arrhythmias, our findings indeed provide new evidence for individualized arrhythmias prevention and treatment in patients with IBD. In addition, the average effect estimate for atrial fibrillation was lower in the present study than in previous studies [16,17], which may be due to the relatively shorter follow-up times in those studies (4.9 [16] and 6.8 [17] years versus 13 years for CD/UC and 9 years for IBD-U in our study). The highest relative risk of arrhythmias was observed shortly after IBD diagnosis, which may be partly explained by detection bias and surveillance bias [6]. Two studies have also investigated the arrhythmia prevalence in patients with IBD [18,19]. However, direct comparison of our results with theirs should be made with caution, as they used different methods of ascertainment for IBD cases and controls, did not validate the diagnosis of IBD or arrhythmias, and had varying degree of adjustment for the potential confounders.

Interestingly, in our study a significantly increased risk was only observed in tachyarrhythmias, although the point estimates for bradyarrhythmias were also greater than one. We speculated that complications (e.g., fistulae and abscesses) and treatment for IBD (e.g., surgery) might play a role here. For example, a 5-fold increased risk of atrial fibrillation/flutter was observed in patients who had undergone surgery [41], and the lifetime risk of surgery ranged from 50% to 80% in patients with CD and reached 30% (colectomy) in patients with UC [42].

Although the underlying mechanisms linking IBD to arrhythmias are intricate, chronic systemic inflammatory activation seems to be the key component [15]. Inflammatory cytokines, particularly tumor necrosis factor (TNF), interleukin(IL)-1, and IL-6, exert arrhythmogenic effects through directly affecting cardiac structural and electrical changes and indirectly affecting the function of other systems (e.g., liver, adipose, and nervous tissue) [14,43]. Moreover, increased C-reactive protein (an inflammatory biomarker) levels have been linked to atrial fibrillation development, recurrence, perpetuation, and thromboembolic complications [44]. Other contributors, as suggested in other CVD studies, might include elevated oxidative stress, platelet and endothelial dysfunction, hypercoagulability, and shifted gut microbiota [45,46]. Similar to IBD, other immune-mediated inflammatory diseases (e.g., rheumatoid arthritis) have also been shown to have a role on the increased risk of arrhythmias [23].

To our knowledge, our study is the first to investigate the associations of IBD with overall and specific arrhythmias in the same population. Strengths include a nationwide cohort design with comprehensive histopathologic data, large sample size, and individually matched references and full siblings, enabling us to perform informative subgroup and sensitivity analyses, to explore the association up to 25 years after IBD diagnosis, and to account for potential arrhythmias risk factors. Our study also benefited from virtually complete follow-up via linkages to national healthcare registers and objective ascertainment of validated IBD and arrhythmias with high PPVs [26,33]. These features minimize selection and information biases commonly seen in observational studies. Sibling comparison helped allay concerns about potential residual confounding due to unmeasured familial factors relevant to IBD and arrhythmias.

We acknowledge some limitations. First, due to lack of data from primary care setting, incomplete coverage of inpatient care data before 1987, and lack of outpatient care data before 2001, the matched references may have included patients with undiagnosed arrhythmias, which might have diluted the real association toward null. However, this concern may be mitigated by the fact that we found no significant difference between different calendar periods. Second, residual confounding is a concern because we lacked complete information on risk

and protective factors for arrhythmias, including lifestyle [47] (e.g., smoking and alcohol consumption) and genetic factors [48]. Nevertheless, the main results were robust after multivariable adjustment and after applying a series of sensitivity analyses. Moreover, such concern may also be partly relieved by the similar results between the population and sibling comparisons, since these factors tend to cluster within families. Similarly, the conservative estimates and wider CIs observed in the sibling comparison analysis, compared to the population comparison analysis, could also be explained by unmeasured confounding shared by siblings (e.g., genetics or early environmental factors) or statistical power issue caused by decreased sample size. Third, ascertainment bias is unlikely to fully explain our results, since the positive association between IBD and arrhythmias persisted even 25 years after diagnosis. Moreover, such concern might be partly alleviated by adjusting for number of healthcare visit (a proxy for regular healthcare seeking behavior) in the analyses and by the similar results obtained after excluding individuals with a healthcare visit between 2 years and 6 months before the index date.

Fourth, since medications (e.g., anti-TNF agents) may decrease inflammatory burden and thereby the risk of CVD (e.g., ischemic heart disease) [8], future studies are warranted to understand how specific IBD medications could influence incidence of arrhythmias. However, due to limited data on IBD medications (especially biologics), potential effects of drug treatment as well as the disentanglement of drug treatment and IBD severity were not within the scope of this paper. Fifth, we lacked data on inflammatory markers, such as C-reactive protein and fecal calprotectin, to assess disease activity. Sixth, given the differences in the incidence and prevalence of IBD [2,4] and arrhythmias [21] between countries, regions, and ethnic groups (ethnicity is not available in our data), and the Swedish healthcare system providing universal access practically free of charge, generalization of our findings to other settings should be cautious. Last, the observational nature of the study restrains from proving a causal link.

Although the observed relative risk of arrhythmias among patients with IBD is small, given its IR among general population is not extremely low (e.g., in those aged ≥60 years: approximately 200 per 10,000 person-years for overall arrhythmias), our findings have important implications. First, clinicians should be aware of the long-term increased risk of arrhythmias in patients with IBD, such as those with other extraintestinal manifestations and those diagnosed ≥60 years of age (in terms of absolute risk), which may help to identify high-risk individuals for arrhythmias at relatively early stage. For those high-risk patients, a long-term follow-up and a risk assessment of modifiable and established CVD risk factors should be considered, since the influence of IBD on arrhythmias is expected to escalate with the increasing burden of CVD risk factors with age. Moreover, previous evidence has suggested that some medications (e.g., steroids) may lead to an increased risk of adverse CVD events, while others (e.g., anti-TNF agents or 5-aminosalicylic acid) may have a cardio-protective effect [9], the optimization of anti-inflammatory therapy while not triggering arrhythmias in those with additional established CVD risk factors should be carefully considered. In addition, there are currently no specific guidelines for the assessment and management of CVD in patients with IBD [8] and the robust findings of the present study underscore the urgency of developing relevant clinical guidelines.

In conclusion, we found that patients with IBD had an elevated risk of arrhythmias for at least 25 years after IBD diagnosis, compared with their matched reference individuals and IBD-free full siblings. Given the rise in IBD prevalence globally [4] (approximately 0.8% Swedish population with IBD in 2017 [6,7]), both patients and clinicians should give special attention regarding the increased arrhythmias or even CVD burden in patients with IBD.

## Supporting information

**S1 Checklist. STROBE Statement—checklist of items that should be included in reports of observational studies.**
(DOC)

**S1 Appendix. Supplementary figures and tables.** Figure A. Hazard ratio (HR) and 95% confidence interval (CI) of specific arrhythmias, comparing inflammatory bowel disease patients with their reference individuals. Figure B. Standardized cumulative incidence and 95% CI of specific arrhythmias in inflammatory bowel disease patients (pink) and their reference individuals (blue). Table A. Previous important studies of inflammatory bowel disease and arrhythmias. Table B. International Classification of Disease (ICD) codes and SNOMED codes defining inflammatory bowel disease. Table C. ICD codes assigned for phenotypes of inflammatory bowel disease. Table D. Definitions of primary and secondary outcomes according to ICD codes. Table E. Definitions of comorbidities according to ICD codes. Table F. Definitions of prescription medications according to ATC codes. Table G. Cumulative incidence difference (95% CI) of arrhythmias during follow-up in individuals with inflammatory bowel disease, compared with their matched reference individuals. Table H. Incident overall arrhythmias in patients with inflammatory bowel disease and their matched reference individuals, stratified by sex, age at index date, calendar period, educational attainment, and number of healthcare visits. Table I. Incident overall arrhythmias in patients with inflammatory bowel disease and their matched reference individuals, stratified by the phenotypes of the Montreal Classification. Table J. Incident specific arrhythmias in patients with inflammatory bowel disease and their matched reference individuals, stratified by the phenotypes of the Montreal Classification. Table K. Sensitivity analyses of the incident arrhythmia in patients with inflammatory bowel disease and their matched reference individuals. Table L. Incident arrhythmia in patients with inflammatory bowel disease and their matched reference individuals (1-year or 3-years lag time). Table M. Characteristics of patients with inflammatory bowel disease and their IBD-free full siblings. Table N. Incident arrhythmia in patients with inflammatory bowel disease and their IBD-free full siblings.
(DOCX)

**S2 Appendix. Flexible parametric survival model vs. Cox regression model.**
(DOCX)

**S1 Text. Analysis plan.**
(DOCX)

## Author Contributions

**Conceptualization:** Jiangwei Sun, Jonas F. Ludvigsson.

**Formal analysis:** Jiangwei Sun.

**Funding acquisition:** Jonas F. Ludvigsson.

**Investigation:** Jiangwei Sun, Jonas F. Ludvigsson.

**Methodology:** Jiangwei Sun, Bjorn Roelstraete, Jonas F. Ludvigsson.

**Project administration:** Jonas F. Ludvigsson.

**Resources:** Jonas F. Ludvigsson.

**Software:** Jiangwei Sun.

**Supervision:** Jonas F. Ludvigsson.

**Visualization:** Jiangwei Sun.

**Writing – original draft:** Jiangwei Sun, Bjorn Roelstraete, Emma Svennberg, Jonas Halfvarson, Johan Sundström, Anders Forss, Ola Olén, Jonas F. Ludvigsson.

**Writing – review & editing:** Jiangwei Sun, Bjorn Roelstraete, Emma Svennberg, Jonas Halfvarson, Johan Sundström, Anders Forss, Ola Olén, Jonas F. Ludvigsson.

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
