## [Editor Report · Decision Letter 0]

26 Apr 2023

Dear Dr Sun, 

Thank you for submitting your manuscript entitled "Long-term risk of arrhythmias in patients with inflammatory bowel disease: A population-based, sibling-controlled cohort study" for consideration by PLOS Medicine.

Your manuscript has now been evaluated by the PLOS Medicine editorial staff and I am writing to let you know that we would like to send your submission out for external peer review.

Please re-submit your manuscript within two working days, i.e. by Apr 28 2023 11:59PM.

Kind regards,

Alex Schaefer, PhD

Associate Editor

PLOS Medicine

---

## [Decision Letter · Decision Letter 1]

27 Jun 2023

Dear Dr. Sun,

Thank you very much for submitting your manuscript "Long-term risk of arrhythmias in patients with inflammatory bowel disease: A population-based, sibling-controlled cohort study" (PMEDICINE-D-23-01095R1) for consideration at PLOS Medicine. 

Your paper was evaluated by an associate editor and discussed among all the editors here. It was also discussed with an academic editor with relevant expertise, and sent to independent reviewers, including a statistical reviewer. The reviews are appended at the bottom of this email and any accompanying reviewer attachments can be seen via the link below:

[LINK]

In light of these reviews, I am afraid that we will not be able to accept the manuscript for publication in the journal in its current form, but we would like to consider a revised version that addresses the reviewers' and editors' comments. Obviously we cannot make any decision about publication until we have seen the revised manuscript and your response, and we plan to seek re-review by one or more of the reviewers. 

We expect to receive your revised manuscript by Jul 18 2023 11:59PM. Please email us (plosmedicine@plos.org) if you have any questions or concerns.

We look forward to receiving your revised manuscript. 

Sincerely,

Alexandra Schaefer, PhD

PLOS Medicine

plosmedicine.org

GENERAL COMMENTS

Please respond to all editor and reviewer comments.

Please cite your Supporting Information as outlined here: https://journals.plos.org/plosmedicine/s/supporting-information

Please ensure that the study is reported according to the STROBE guideline, and include the completed STROBE checklist as Supporting Information. Please add the following statement, or similar, to the Methods: "This study is reported as per the Strengthening the Reporting of Observational Studies in Epidemiology (STROBE) guideline (S1 Checklist)."

Did your study have a prospective protocol or analysis plan? Please state this (either way) early in the Methods section.

For all observational studies, in the manuscript text, please indicate: (1) the specific hypotheses you intended to test, (2) the analytical methods by which you planned to test them, (3) the analyses you actually performed, and (4) when reported analyses differ from those that were planned, transparent explanations for differences that affect the reliability of the study's results. If a reported analysis was performed based on an interesting but unanticipated pattern in the data, please be clear that the analysis was data-driven.

EDITORIAL COMMENTS

During the internal discussion of your manuscript, we noted that the increased risk of arrhythmias appears to be attributed to undiagnosed CVD. However, there does not seem to be any evidence from the data presented that the increased risk of arrhythmias is vascular in origin, e.g. triggered by coronary artery disease due to steroid use, as implied in your cover letter. Steroids are most commonly associated with bradyarrhythmias, but in your study you report tachyarrhythmias in the study population. In addition, as you report, e.g., due to possible intestinal surgeries/stomas, disease severity seems to be as important as disease activity (i.e. outside of the acute illness). GI surgeries/stoma complications often lead to acute and chronic electrolyte disturbances, which in turn disrupt cardiac electrophysiology. In addition, many systemic treatments for IBD (other than steroids) cause renal disease of a tubular nature, which significantly alters electrolyte balance. Suggest framing your discussion/implications of your findings to reflect this including implications for long term care/follow-up/monitoring.

ABSTRACT

Abstract Background:

*Provide the context of why the study is important. The final sentence should clearly state the study question.

Please ensure that all numbers presented in the abstract are present and identical to numbers presented in the main manuscript text.

PLOS Medicine requests that main results are quantified with 95% CIs as well as p values. Please include. When reporting p values please report as p<0.001 and where higher as the exact p value p=0.002, for example. For the purposes of transparent data reporting, if not including the aforementioned please clearly state the reasons why not.

Please include any important dependent variables that are adjusted for in the analyses.

Throughout, suggest reporting statistical information as follows to improve clarity for the reader “22% (95% CI [13%,28%]; p</=)”. Please amend throughout the abstract and main manuscript.

Please note the use of commas to separate upper and lower bounds, as opposed to hyphens as these can be confused with reporting of negative values.

When a p value is given, please specify the statistical test used to determine it.

Please provide brief demographic details of the study population (e.g., sex, age, ethnicity, etc.).

Abstract Conclusions:

*Please begin your Abstract Conclusions with "In this study, we observed ..." or similar, to summarize the main findings from your study, without overstating your conclusions. Please emphasize what is new and address the implications of your study, being careful to avoid assertions of primacy. 

*Please avoid vague statements such as ""these results have major implications for policy/clinical care"". Mention only specific implications substantiated by the results.

AUTHOR SUMMARY

At this stage, we ask that you include a short, non-technical Author Summary of your research to make findings accessible to a wide audience that includes both scientists and non-scientists. The Author Summary should immediately follow the Abstract in your revised manuscript. This text is subject to editorial change and should be distinct from the scientific abstract. Please see our author guidelines for more information: https://journals.plos.org/plosmedicine/s/revising-your-manuscript#loc-author-summary.

The summary should include 2-3 single sentence, individual bullet points under each of the questions.

It may be helpful to review currently published articles for examples which can be found on our website here https://journals.plos.org/plosmedicine/

INTRODUCTION

l.34: Please change ‘western’ to ‘Western’.

l.92: Please include a reference at the end of the statement (‘including stroke, ischemic heart disease, and venous thromboembolism event’). 

Please address past research and explain the need for and potential importance of your study. Indicate whether your study is novel and how you determined that. If there has been a systematic review of the evidence related to your study (or you have conducted one), please refer to and reference that review and indicate whether it supports the need for your study. 

Please conclude the Introduction with a clear description of the study question or hypothesis.

METHODS AND RESULTS

PLOS Medicine requests that main results are quantified with 95% CIs as well as p values. Please include. When reporting p values please report as p<0.001 and where higher as the exact p value p=0.002, for example. For the purposes of transparent data reporting, if not including the aforementioned please clearly state the reasons why not.

Please include any important dependent variables that are adjusted for in the analyses.

Suggest reporting statistical information as detailed above – see under ABSTRACT

Please present numerators and denominators for percentages, at least in the Tables [not necessarily each time they're mentioned].

Please provide brief demographic details of the study population (e.g., sex, age, ethnicity, etc.).

l.154: Please change ‘or December 31, 2019’ to ‘or 31 December 2019’.

l.158: Please define ‘PPV’.

l.167: Please write ‘y’ out in full (year/s) or introduce the abbreviation at first use.

ll.173-174: Please change ‘Data on cardiovascular related comorbidities preceding index date was also collected from the NPR, […]’ to ‘Data on cardiovascular-related comorbidities preceding the index date were also collected from the NPR, […]’.

ll.196-197: For clarity, please change ‘[…] (<18, 18-<40, 40-<60, and ≥60), […]’ to ‘[…] (<18, 18-39, 40-59, and ≥60), […]’.

l.198: Please write ‘y’ out in full (year/s) or introduce the abbreviation at first use.

ll.207-210 suggest: “Third, given that the Prescribed Drug Register became available in July 2005, we restricted the analysis to individuals with index date later than 1 January 2006, and constructed models that further adjusted for the above-mentioned cardiovascular-related medications.”

l.215: Please change ‘potential influence due to confounding’ to ‘potential influence of confounding’.

ll.223-225: Under the subheading ‘Ethics consideration’, please include the information on participant consent presented in lines 141-142. In this way, readers will be able to find all information on ethical requirements in one place.

ll.228-230: “We identified 24,954 CD cases (median age at index date: 38.4 years; female: 52.2%), 46,856 UC cases (42.1 years; 46.3%), and 12,067 IBD-U cases (43.8 years; 49.6%), together with their matched reference individuals (Table 1).” Please ensure that all parentheses indicate that the number of years is the median age at the index date and the percentage is the percentage of female participants.

ll.231-233: Please change to “Compared with reference individuals, patients with IBD had a higher number of healthcare visits and a higher prevalence of previous diseases, including ischemic heart disease, heart failure, stroke, hypertension, diabetes, obesity, dyslipidemia, chronic kidney disease, and COPD.” In addition, please include a reference at the end of this statement to where this information can be found in your manuscript.

ll.239-242: Please ensure that all parentheses indicate the unit and definition of the numbers presented. It is not sufficient to put all the details in one (the first) parenthesis of the sentence and leave the complete details out of the following parentheses. Please check and revise throughout your entire manuscript. 

DISCUSSION

Please remove all subheadings from your Discussion.

ll.334-335: Please temper assertions of primacy ("Our study is the first one to date…”) by adding ‘to the best of our knowledge’ or similar.

FIGURES

For all Figures, please ensure that you have complied with our figures requirements http://journals.plos.org/plosmedicine/s/figures.

For all Figures, including those in Supporting Information files, please show the axis beginning at zero. If this is not possible, please show a break in the axis.

Please define abbreviations used in the figure legend of each figure (including those in Supporting Information files).

Figure 1: Please define ‘IBD’ in the figure description at first use. Also, please define ‘CD’, ‘UC’ and ‘IBD-U’ in the figure description or in a figure legend.

TABLES

Please note the use of commas to separate upper and lower bounds, as opposed to hyphens as these can be confused with reporting of negative values.

Please define abbreviations used in the table description of each table (including those in Supporting Information files).

Table 1: Please define ‘y’ in the abbreviations listed below the table or change the heading to ‘Educational attainment, years’ and remove the ‘y’.

Table 1: For clarity, please change the groups for ‘Age at index date’ from <18, 18-<40, 40-<60, and ≥60 to <18, 18-39, 40-59, and ≥60.

Table 2: Please define ‘Pys’ in in the abbreviations listed below the table.

Table 2: Please write ‘COPD’ out in full.

Supplemental Table 1: Please define ‘CVD’.

Supplemental Table 2: Please define ‘SNOMED’.

Supplemental Table 3: Please define ‘ICD’, ‘PSC’, ‘PI’, ‘L’ and ‘E’.

Supplemental Table 3: Please remove the second comma after ‘M461,,’

Supplemental Table 3: RE reference of Professor Annika Bergquist, please place the personal communication in the text. Please provide the name of the individual, the affiliation, and date of communication. The individual named must provide written permission to be named.

Supplemental Table 4: Please define ‘ICD’.

Supplemental Table 5: Please define ‘ICD’, ‘ATC’.

Supplemental Table 5: It appears you wrongfully cited Supplemental Table 4 as reference for medication ATC codes (‘Medication ATC codes: see antihypertensive medications in Supplemental Table 4’; ‘Medication ATC codes: see antidiabetic medications in Supplemental Table 4’). Instead, Supplemental Table 6 should be cited. 

Supplemental Table 7: Please define ‘COPD’.

Supplemental Table 8: For clarity, please change the groups for ‘Age at index date, years’ from <18, 18-<40, 40-<60, and ≥60 to <18, 18-39, 40-59, and ≥60. Also, please define ‘y’ in the abbreviations listed below the table or change the heading to ‘Educational attainment, years’ and remove the ‘y’.

Supplemental Table 8: Please define ‘Pys’ and ‘COPD’.

Supplemental Table 9: Please define ‘COPD’, ‘L’ and ‘E’.

Supplemental Table 10: Please define ‘COPD’, ‘L’ and ‘E’.

Supplemental Table 11: Please define ‘COPD’ and remove the definition for ‘IR’ as it is not part of the table.

Supplemental Table 12: Please define ‘COPD’.

Supplemental Table 13: For clarity, please change the groups for ‘Age at index date, years’ from <18, 18-<40, 40-<60, and ≥60 to <18, 18-39, 40-59, and ≥60. Also, please define ‘y’ in the abbreviations listed below the table or change the heading to ‘Educational attainment, years’ and remove the ‘y’.

Supplemental Table 14: Please define ‘COPD’ and ‘Pys’.

REFERENCES

PLOS uses the numbered citation (citation-sequence) method and first six authors, et al.

Please ensure that journal name abbreviations match those found in the National Center for Biotechnology Information (NCBI) databases (http://www.ncbi.nlm.nih.gov/nlmcatalog/journals), and are appropriately formatted and capitalised.

Please also see https://journals.plos.org/plosmedicine/s/submission-guidelines#loc-references for further details on reference formatting. 

Comments from the reviewers:

Reviewer #1: I would like to thank the authors for this interesting work on the risk of cardiac arrhythmias in IBD patients. The large number of patients included, the long follow-up period, and the use of a controlled matched sibling population support the results of this study despite its retrospective nature. As correctly explained in the text, the major biases include the lack of an assessment of IBD activity and the lack of details on IBD treatments, both of which have an important impact on the risk of arrhythmias. Nevertheless, the text is well written, and only minor revisions are suggested: 

Introduction Chapter:

- Line 88, please replace "have" with "has".

- Line 101-103. there is evidence of ventricular arrhythmias (Please add: Mubasher M et al. An Investigation into the Association Between Inflammatory Bowel Disease and Cardiac Arrhythmias: An Examination of the United States National Inpatient Sample Database. Clin Med Insights Cardiol. 2020. PMID: 33192109). Please cite it in the text with a brief description.

- Line 112. reference 8 "Sinh P et al. Cardiovascular Risk Assessment and Impact of Medications on Cardiovascular Disease in Inflammatory Bowel Disease" is a literature review and should not be used to confirm the lack of specific guidelines. Please remove. 

Chapter Results:

- The results were correctly adjusted for several variables associated with a higher risk of cardiac arrhythmias. Among them, thyroid dysfunction (both hypo- and hyperthyroidism) was not mentioned. Since the European prevalence of hypothyroidism and hyperthyroidism is about 5% (Please add: Chiovato L, et al. Hypothyroidism in Context: Where We have Been and Where We are Going. Adv Ther. 2019 PMID: 31485975) and 1% (Please add: Caputo M, et al. Incidence and prevalence of hyperthyroidism: a population-based study in the Piedmont region, Italy. Endocrine. 2020. PMID: 32056093). If available, results should also be adjusted for this parameter. 

Discussion chapter: 

- Mechanism section: please write a few words about the arrhtymogenic properties of TNF, IL -1 and IL -6 (Please add: Massironi S, et al. The often overlooked cardiovascular complications of inflammatory bowel disease. Expert Rev Clin Immunol. 2023 PMID: 36722283). This is indeed an important message for future studies on the arrhythmogenic effects of specific immunosuppressants.

- According to this work, there is a category of IBD patients at higher risk for cardiac arrhythmias (Chapter Discussion, Implications section", lines 402-405). Despite the lack of information on disease activity and treatment, this important message for clinicians should be presented graphically in a simple and clear picture. 

Reviewer #2: I read with interest this study, that confirms and extend what previously known from researches on atrial fibrillation in IBD. The study design is robust and articulated and the Authors correctly acknowledged the limitations of this kind of population studies based on administrative data (i.e. lack of detail on clinical and laboratory parameters of inflammation). It would be important, as suggested, to further expand the research to the analysis of drug prescriptions to verify the hypothesis of a deleterious effect of corticosteroids and a protective one of anti-TNF on the risk of arrythmias in IBD patients.

Reviewer #3: This is a well-conducted population-based cohort study on the long-term risk of arrhythmias in patients with inflammatory bowel disease. The study design, datasets, statistical methods and analyses, and presentation (tables and figures) and interpretation of the results are mostly adequate and of a good standard. However, they are still a couple of issues needing attention.

1) Interpretation of the results. It says in the sibling comparison section on page 21 " Consistent with the primary analysis, UC or IBD-U patients demonstrated significantly higher risk of developing overall arrhythmias [aHR=1.18 (1.09-1.28) for UC; aHR=1.19 (0.99-1.42) for IBD-U]...". However, this is incorrect. Actually the results from reference comparison (Table 2) is quite different from that from sibling comparison (sTable14). The elevated risk of arrhythmia in patients with inflammatory bowel disease as compared to the reference individuals were not generally seen in sTable14 when compared to siblings apart from UC group only (with 2 of 4 subgroups still showing non-significant results). It's intriguing that the increased risk was not seen in the sibling comparison, which probably hints that risk of arrhythmia is not associated with the IBD? If one goes into the details of "Disease history before index date" sections in both Table 2 and sTable14, one can see people with IBD have higher cardiovascular risk factors even before the index IBD, and the elevated risk is more evident in the reference comparison than in the sibling comparison. Then this raises the question: are these risk factors prior to IBD or IBD itself which actually lead to the arrahthmias? Basically the sibling results make the current arguement/conclusion a bit weak.

2) It is adequate to use flexible parametric survival models in the analyses. However, it would be good to write a short paragragh to introduce/explain in a bit details on why a flexible parametric survival model was chosen and how it compares to the well-known Cox model, similarity and difference and etc. The paragraph can be added as supplementary information.

Reviewer #4: The authors take on a highly relevant subject that which has received insufficient attention in the past. They apparently had access to a large cohort that allows for advanced statistics. This paper will help better understand intrinsic cardiovascular risks in IBD.

I have a few concerns:

a) although Northern Europeans have a distinct understanding of Nordic countries, this could be confusion, because S, NO, FIN and DK are not the only Nordic countries, the Baltic countries would meet that definition as well let alone Canada and Russia etc. I would define this somewhere.

b) Even within S, NO, FIN and DK, not all people share the same ethnicity. I am not talking about migrants and immigrants here but the native communities in Greenland or Sámi peoples.

c) I am not sure I fully understand how medications that have a profound impact on CV risk were considered? For instance, 5-ASA is a compound similar to ASA which has preventive effects of MACE and VTE. Please elaborate on that.

d) And yes, it is a major limitation that the confounding effect of active vs. inactive disease disease could not be considered. This should be made more clear in the abstract 

Reviewer #5: I have no issues with the analytic methods of the study With aHR of 1.13, 1.14 the authors need to discuss that considering case ascertainment bias by having more health care visits and that COPD contacts is not a great proxy for all smoking and since UC and CD patients are both more highly likely to have ever been smokers perhaps this is all that is being seen..its not much of a risk

[LINK]

---

## [Decision Letter · Decision Letter 2]

12 Sep 2023

Dear Dr. Sun,

Thank you very much for re-submitting your manuscript "Long-term risk of arrhythmias in patients with inflammatory bowel disease: A population-based, sibling-controlled cohort study" (PMEDICINE-D-23-01095R2) for review by PLOS Medicine.

I have discussed the paper with my colleagues and the academic editor and it was also seen again by two reviewers. I am pleased to say that provided the remaining editorial and production issues are dealt with we are planning to accept the paper for publication in the journal.

[LINK]

We look forward to receiving the revised manuscript by Sep 19 2023 11:59PM.   

Sincerely,

Alexandra Schaefer, PhD 

Associate Editor 

PLOS Medicine

plosmedicine.org

Requests from Editors:

GENERAL COMMENTS

Thank you for considered and detailed responses to editor and reviewer comments.

Please see below for further minor points that we request you respond to in full.

Please revise your reporting of statistical information throughout the entire manuscript (see below for details). It is important to use a consistent and clear format for the reader. 

Please check your manuscript carefully for grammar, spelling and punctuation. 

Please replace “IBD patients” with “patients with IBD” throughout your manuscript. 

Please leave no space between the superscript (note number) in the text and the word, number or mark of punctuation it follows. Please revise throughout the entire manuscript (including the Supplementary information files).

FINANCIAL DISCLOSURE

The funding statement should include: specific grant numbers, initials of authors who received each award, URLs to sponsors’ websites. 

EDITORIAL COMMENTS 

We initially suggested framing your discussion/implications of your findings to reflect implications of your study for long-term care/follow-up/monitoring. We feel this has not been sufficiently addressed and would ask you to consider these aspects in your discussion.

ABSTRACT

ll.59-61: Please use a consistent format when reporting statistical information (including consecutive brackets). We suggest reporting statistical information as follows to improve clarity for the reader “22% (95% CI [13%,28%]; p</=)”. Please amend throughout the abstract and main manuscript. 

Editorial suggestion: “Compared to reference individuals, overall arrhythmias were increased in CD [54.6 vs. 46.1 per 10,000 person-years; aHR=1.15 (95% CI [1.09, 1.21]; P<0.001]), UC [64.7 vs. 53.3 per 10,000 person-years; aHR=1.14 (95% CI [1.10, 1.18]; P<0.001), and IBD-U patients [78.1 vs. 53.5 per 10,000 person-years; aHR=1.30 (95% CI [1.20, 1.41]; P<0.001).”

AUTHOR SUMMARY

Please revise second subheading to ‘What Did the Researchers Do and Find?’, and describe the methods of the study in non-technical language.

Thank you for providing the Author Summary. In the final bullet point of ‘What Do These Findings Mean?’, please describe the main limitations of the study in non-technical language.

Please write ‘CVD’ in full words.

METHODS AND RESULTS

Thank you for providing the pre-specified analysis plan in the Methods section and providing the plan as a supplementary file (S1 Text). Please make sure that the Methods section transparently describes when analyses were planned, and when/why any data-driven changes to analyses took place. Changes in the analysis, including those made in response to peer review comments, should be identified as such in the Methods section of the paper, with rationale.

l.265 suggest: “About 14.6% of CD cases were colonic and 15.5% of UC cases had extensive colitis (Table 1).”

ll.270-274: For clarity and easier comparison, we suggest reporting the percentages for the CD/UC/IBD-cases and including the actual case numbers in parenthesis (i.e. switching percentages with case numbers).

ll.275-278: Please revise the reporting of the statistical details (see above, example: “with an average aHR of 1.15 (95% CI [1.09, 1.21], P<0.001) for CD)”). Please revise throughout the entire manuscript.

ll.303-308: Please report exact p-values.

ll.311-314/ll.319-324: Please report the full statistical details. Please revise throughout the entire manuscript.

l.314: Please revise “non-significantly increased risk of bradyarrhythmias” to “There was no evidence of an significant association between IBD and bradyarrythmias” or similar.

ll.342-343: Please report exact p-values.

DISCUSSION

When discussing other studies, you gave the percentage of male participants (e.g., l.359). In your study, you consistently reported the percentage of females. We suggest changing the information in the discussion for consistency and clarity for the reader.

ll.432-437, please change to: “Fourth, since medications (e.g., anti-TNF agents) may decrease inflammatory burden and thereby the risk of CVD (e.g., ischemic heart disease) [8], future studies are warranted to understand how specific IBD medications could influence incidence of arrhythmias. However, due to limited data on IBD medications (especially biologics), potential effects of drug treatment as well as the disentanglement of drug treatment and IBD severity were not within the scope of this paper."

ll.438-441, please change to: Sixth, given the differences in the incidence and prevalence of IBD [2,4] and arrhythmias [21] between countries, regions, and ethnic groups (ethnicity is not available in our data), and the Swedish health care system providing universal access practically free of charge, generalization of our findings to other settings should be cautious.

l.445, please change to: “those aged ≥ 60 years”.

ll.447-450, please change to: First, clinicians should be aware of the long-term increased risk of arrhythmias in IBD patients, such as those with other extraintestinal manifestations and those diagnosed at ≥60 years of age (in terms of absolute risk), which may help to identify high-risk individuals for arrhythmias at a relatively early stage.

ll.456-458, please change to: In addition, there are currently no specific guidelines for the assessment and management of CVD in IBD patients [8], and the robust findings of the present study underscore the urgency of developing relevant clinical guidelines.

ll.461-462: “Given the rise in IBD prevalence globally [4] (~0.8% Swedish population with IBD in 2017) [6,7], […]” – It appears reference 6 and 7 should be included in the parenthesis, please revise.

REFERENCES

Please revise your references thoroughly. For example, in the first reference “The New England journal of medicine” should be abbreviated as “N Engl J Med”.

PLOS uses the numbered citation (citation-sequence) method and first six authors, et al. 

Please ensure that journal name abbreviations match those found in the National Center for Biotechnology Information (NCBI) databases (http://www.ncbi.nlm.nih.gov/nlmcatalog/journals), and are appropriately formatted and capitalised. Please also see https://journals.plos.org/plosmedicine/s/submission-guidelines#loc-references for further details on reference formatting. Where website addresses are cited, please specify the date of access. 

FIGURES

Figure 1: Currently, ‘CD’, ‘UC’ and ‘IBD-U’ could be mistaken for the y-axis title. For clarity, we suggest re-arranging the graph (from 3 rows/2 columns to 2 rows/3 columns). We suggest that the disease conditions function as titles for the 3 columns of graphs and that “HR (95% CI)” and “Cumulative incidence (%)” move to the left as y-axis titles (2 rows of graphs). The “years since index date” as the x-axis title should appear under each column of graphs. Subsequently, in the figure description you should change “left” and “right” to “upper row” and “lower row” or mark the two rows of graphs as “A” and “B” and describe this in the figure description accordingly.

Figure A: Please add “HR (95% CI)” to each y-axis on the left and ensure that the y-axis title is right next to y-axis (so that ‘CD’, ‘UC’ and ‘IBD-U’ do not get mistaken for the y-axis title). The x-axis title “years since index date” should appear under each column of graphs. Please ensure consistency in the number format on the y-axis (one decimal point versus none).

Figure B: Please adjust similarly to Figure A.

TABLES

Table 1: We note that in parentheses you provide the percentage of, for example, under 18 years old CD cases (11.5) without clarifying the meaning of the numbers in the parentheses (i.e. a unit is missing). Please revise. We suggest changing the column header to ‘XY cases/references, n (%)’ or similar (as done in Table 2).

Table 1: In line with Reviewer 4’s comment, we suggest adding a footnote defining ‘Nordic countries’.

Table A: For the columns “Follow-up time” and “Mean age”, please add a unit (e.g., years). Please define ‘US’ and ‘USA’. In the column “Main findings”, please ensure to report statistical information correctly and fully (e.g., mention 95% CI). 

Table A: The country listed for “Population-based cohort, Choi, 2019” should be Korea (not Korean).

Table B: Please change “(i.g., “M…”)” to “(e.g., “M…”).

Table D: We suggest changing the title to “Definitions of primary and secondary outcomes according to ICD codes”.

Table E: Please change “only if patient diagnosed ≥40 year” to “only if patient diagnosed ≥40 years old”.

Table E: We suggest changing the title to “Definitions of comorbidities according to ICD codes”.

Table F: We suggest changing the title to “Definitions of prescription medications according to ATC codes” and changing the second column header to “Definition” in line with previous tables.

Table I/J/L: We suggest adjusting the table format avoiding unnecessary line breaks to enhance accessibility (e.g., please avoid a line break in the word “References”).

Table M: In line with Reviewer 4’s comment, we suggest adding a footnote defining ‘Nordic countries’.

Table M: Please see comment Table 1 (missing details/units).

SUPPLEMENTARY MATERIAL

S1 Text: Please ensure that in the Statistical Analysis Plan provided all abbreviations including those in the tables are defined and introduced at first use.

S2 Appendix: In the first sentence, please change ‘model’ to ‘models’. Please also change “Compared to Cox regression model, flexible parametric survival model” to “Compared with the Cox regression model, flexible parametric survival models”.

SOCIAL MEDIA

To help us extend the reach of your research, please provide any Twitter handle(s) that would be appropriate to tag, including your own, your coauthors’, your institution, funder, or lab. Please respond to this email with any handles you wish to be included when we tweet this paper.

Comments from Reviewers:

Reviewer #1: The paper has substantially improved and can be accepted

Reviewer #3: Thanks authors for their great effort to improve the manuscript. I am satisfied with the response and revision. No further issues needing attention.

[LINK]

---

## [Editor Report · Decision Letter 3]

2 Oct 2023

Dear Dr Sun, 

On behalf of my colleagues and the Academic Editor, Sanjay Basu, I am pleased to inform you that we have agreed to publish your manuscript "Long-term risk of arrhythmias in patients with inflammatory bowel disease: A population-based, sibling-controlled cohort study" (PMEDICINE-D-23-01095R3) in PLOS Medicine.

I would like to thank you for your considered and detailed responses to the editorial comments. We are excited to publish your manuscript, and there are some remaining minor stylistic and presentation points that should be addressed prior to publication. We will carefully review whether these changes have been made. If there are any questions concerning these final requests, please do not hesitate to contact me at aschaefer@plos.org. 

Please see below for further minor points that we request you respond to:

1) Please change the short title to “Long-term risk of arrhythmia in patients with IBD”

2) Methods: We appreciate that you have identified data-driven changes to analyses in the Methods section of the paper with rationale. However, in the main manuscript we feel it is not clear to the reader that these changes were made in response to peer review comments (only in the S1 Text this becomes clear). Please add a sentence explaining that the sensitivity analyses mentioned under second and fourth have been added during the peer-review process and are non-prespecified analyses.

3) l.308 (description Figure 1) “inflammatory bowel disease (IBD) patients” – Please change to “patients with inflammatory bowel disease (IBD)”. Please check carefully through your entire manuscript including figure/table descriptions. Please also adjust accordingly any mentions of ‘CD patients’, ‘UC patients’ and ‘IBD-U patients’ (and others).

4) Please use a consistent format of round and/or square brackets when reporting statistical information. For example, l.330 “(aHR=1.20 (95% CI [1.02, 1.41], P<0.001)” versus l.349 “[aHR=2.22 (95% CI: 1.40, 3.52)]”. Please check throughout the main manuscript.

PRESS

Sincerely, 

Alexandra Schaefer, PhD 

Associate Editor 

PLOS Medicine